# Learning Generative Models with Visual Attention

Yichuan Tang,  Nitish Srivastava,  Ruslan Salakhutdinov

Department of Computer Science
University of Toronto
Toronto, Ontario, Canada
{tang,nitish,rsalakhu}@cs.toronto.edu

## Abstract

Attention has long been proposed by psychologists to be important for efficiently dealing with the massive amounts of sensory stimulus in the neocortex. Inspired by the attention models in visual neuroscience and the need for object-centered data for generative models, we propose a deep-learning based generative framework using attention. The attentional mechanism propagates signals from the region of interest in a scene to an aligned canonical representation for generative modeling. By ignoring scene background clutter, the generative model can concentrate its resources on the object of interest. A convolutional neural net is employed to provide good initializations during posterior inference which uses Hamiltonian Monte Carlo. Upon learning images of faces, our model can robustly attend to the face region of novel test subjects. More importantly, our model can learn generative models of new faces from a novel dataset of large images where the face locations are not known.

## 1  Introduction

Building rich generative models that are capable of extracting useful, high-level latent representations from high-dimensional sensory input lies at the core of solving many AI-related tasks, including object recognition, speech perception and language understanding. These models capture underlying structure in data by defining flexible probability distributions over high-dimensional data as part of a complex, partially observed system. Some of the successful generative models that are able to discover meaningful high-level latent representations include the Boltzmann Machine family of models: Restricted Boltzmann Machines, Deep Belief Nets [1], and Deep Boltzmann Machines [2]. Mixture models, such as Mixtures of Factor Analyzers [3] and Mixtures of Gaussians, have also been used for modeling natural image patches [4]. More recently, denoising auto-encoders have been proposed as a way to model the transition operator that has the same invariant distribution as the data generating distribution [5].

Generative models have an advantage over discriminative models when part of the images are occluded or missing. Occlusions are very common in realistic settings and have been largely ignored in recent literature on deep learning. In addition, prior knowledge can be easily incorporated in generative models in the forms of structured latent variables, such as lighting and deformable parts. However, the enormous amount of content in high-resolution images makes generative learning difficult [6, 7]. Therefore, generative models have found most success in learning to model small patches of natural images and objects: Zoran and Weiss [4] learned a mixture of Gaussians model over $8\times8$ image patches; Salakhutdinov and Hinton [2] used $64\times64$ centered and uncluttered stereo images of toy objects on a clear background; Tang et al. [8] used $24\times24$ images of centered and cropped faces. The fact that these models require curated training data limits their applicability on using the (virtually) unlimited unlabeled data.

In this paper, we propose a framework to infer the region of interest in a big image for generative modeling. This will allow us to learn a generative model of faces on a very large dataset of (unlabeled) images containing faces. Our framework is able to dynamically route the relevant information to the generative model and can ignore the background clutter. The need to dynamically and selectively route information is also present in the biological brain. Plethora of evidence points to

the presence of attention in the visual cortex [9, 10]. Recently, in visual neuroscience, attention has been shown to exist not only in extrastriate areas, but also all the way down to V1 [11].

Attention as a form of routing was originally proposed by Anderson and Van Essen [12] and then extended by Olshausen et al. [13]. Dynamic routing has been hypothesized as providing a way for achieving shift and size invariance in the visual cortex [14, 15]. Tsotsos et al. [16] proposed a model combining search and attention called the Selective Tuning model. Larochelle and Hinton [17] proposed a way of using third-order Boltzmann Machines to combine information gathered from many foveal glimpses. Their model chooses where to look next to find locations that are most informative of the object class. Reichert et al. [18] proposed a hierarchical model to show that certain aspects of *covert* object-based attention can be modeled by Deep Boltzmann Machines. Several other related models attempt to learn where to look for objects [19, 20] and for video based tracking [21]. Inspired by Olshausen et al. [13], we use 2D similarity transformations to implement the scaling, rotation, and shift operation required for routing. Our main motivation is to enable the learning of generative models in big images where the location of the object of interest is unknown a-priori.

## 2   Gaussian Restricted Boltzmann Machines

Before we describe our model, we briefly review the Gaussian Restricted Boltzmann Machine (GRBM) [22], as it will serve as the building block for our attention-based model. GRBMs are a type of Markov Random Field model that has a bipartite structure with real-valued visible variables $\mathbf{v} \in \mathbb{R}^D$ connected to binary stochastic hidden variables $\mathbf{h} \in \{0, 1\}^H$. The energy of the joint configuration $\{\mathbf{v}, \mathbf{h}\}$ of the Gaussian RBM is defined as follows:

$$E_{GRBM}(\mathbf{v}, \mathbf{h}; \Theta) \quad = \quad \frac{1}{2} \sum_i \frac{(v_i - b_i)^2}{\sigma_i^2} - \sum_j c_j h_j - \sum_{ij} W_{ij} v_i h_j, \tag{1}$$

where $\Theta = \{\mathbf{W}, \mathbf{b}, \mathbf{c}, \boldsymbol{\sigma}\}$ are the model parameters. The marginal distribution over the visible vector $\mathbf{v}$ is $P(\mathbf{v}; \Theta) = \frac{1}{\mathcal{Z}(\Theta)} \sum_{\mathbf{h}} \exp\left(-E(\mathbf{v}, \mathbf{h}; \Theta)\right)$ and the corresponding conditional distributions take the following form:

$$p(h_j = 1 | \mathbf{v}) \quad = \quad 1/\left(1 + \exp(-\sum_i W_{ij} v_i - c_j)\right), \tag{2}$$

$$p(v_i | \mathbf{h}) \quad = \quad \mathcal{N}(v_i; \mu_i, \sigma_i^2), \quad \text{where} \quad \mu_i = b_i + \sigma_i^2 \sum_j W_{ij} h_j. \tag{3}$$

Observe that conditioned on the states of the hidden variables (Eq. 3), each visible unit is modeled by a Gaussian distribution, whose mean is shifted by the weighted combination of the hidden unit activations. Unlike directed models, an RBM's conditional distribution over hidden nodes is factorial and can be easily computed.

We can also add a binary RBM on top of the learned GRBM by treating the inferred $\mathbf{h}$ as the "visible" layer together with a second hidden layer $\mathbf{h}^2$. This results in a 2-layer Gaussian Deep Belief Network (GDBN) [1] that is a more powerful model of $\mathbf{v}$.

Specifically, in a GDBN model, $p(\mathbf{h}^1, \mathbf{h}^2)$ is modeled by the energy function of the 2nd-layer RBM, while $p(\mathbf{v}^1 | \mathbf{h}^1)$ is given by Eq. 3. Efficient inference can be performed using the greedy approach of [1] by treating each DBN layer as a separate RBM model. GDBNs have been applied to various tasks, including image classification, video action and speech recognition [6, 23, 24, 25].

## 3   The Model

Let $\mathcal{I}$ be a high resolution image of a scene, e.g. a $256 \times 256$ image. We want to use attention to propagate regions of interest from $\mathcal{I}$ up to a canonical representation. For example, in order to learn a model of faces, the canonical representation could be a $24 \times 24$ aligned and cropped frontal face image. Let $\mathbf{v} \in \mathbb{R}^D$ represent this low resolution canonical image. In this work, we focus on a Deep Belief Network[1] to model $\mathbf{v}$.

This is illustrated in the diagrams of Fig. 1. The left panel displays the model of Olshausen et.al. [13], whereas the right panel shows a graphical diagram of our proposed generative model with an attentional mechanism. Here, $\mathbf{h}^1$ and $\mathbf{h}^2$ represent the latent hidden variables of the DBN model, and

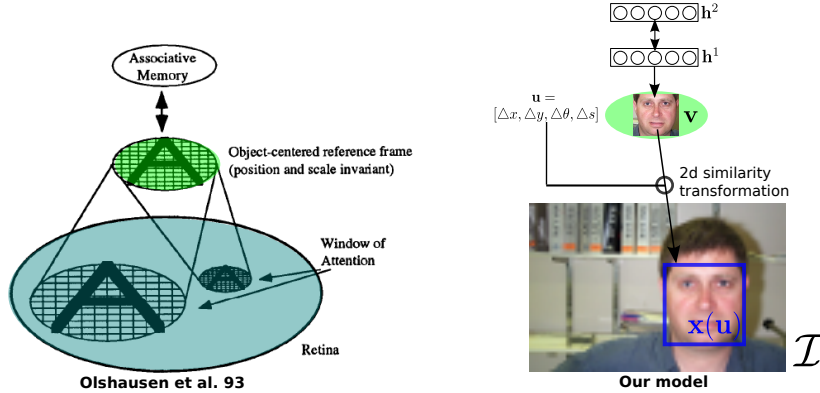

Figure 1: **Left:** The Shifter Circuit, a well-known neuroscience model for visual attention [13]; **Right**: The proposed model uses 2D similarity transformations from geometry and a Gaussian DBN to model canonical face images. Associative memory corresponds to the DBN, object-centered frame correspond to the visible layer and the attentional mechanism is modeled by 2D similarity transformations.

$\triangle x, \triangle y, \triangle \theta, \triangle s$ (position, rotation, and scale) are the parameters of the 2D similarity transformation.

The 2D similarity transformation is used to rotate, scale, and translate the canonical image $\mathbf{v}$ onto the canvas that we denote by $\mathcal{I}$. Let $\mathbf{p} = [x\, y]^\mathsf{T}$ be a pixel coordinate (e.g. $[0, 0]$ or $[0, 1]$) of the canonical image $\mathbf{v}$. Let $\{\mathbf{p}\}$ be the set of all coordinates of $\mathbf{v}$. For example, if $\mathbf{v}$ is 24×24, then $\{\mathbf{p}\}$ ranges from $[0, 0]$ to $[23, 23]$. Let the "gaze" variables $\mathbf{u} \in \mathbb{R}^4 \equiv [\triangle x, \triangle y, \triangle \theta, \triangle s]$ be the parameter of the Similarity transformation. In order to simplify derivations and to make transformations be linear w.r.t. the transformation parameters, we can equivalently redefine $\mathbf{u} = [a,\, b,\, \triangle x,\, \triangle y]$, where $a = s\sin(\theta) - 1$ and $b = s\cos(\theta)$ (see [26] for details). We further define a function $\mathsf{w} := \mathsf{w}(\mathbf{p}, \mathbf{u}) \to \mathbf{p}'$ as the transformation function to *warp* points $\mathbf{p}$ to $\mathbf{p}'$:

$$\mathbf{p}' \triangleq \left[ \begin{array}{c} x' \\ y' \end{array} \right] = \left[ \begin{array}{cc} 1+a & -b \\ b & 1+a \end{array} \right] \left[ \begin{array}{c} x \\ y \end{array} \right] + \left[ \begin{array}{c} \triangle x \\ \triangle y \end{array} \right]. \tag{4}$$

We use the notation $\mathcal{I}(\{\mathbf{p}\})$ to denote the bilinear interpolation of $\mathcal{I}$ at coordinates $\{\mathbf{p}\}$ with anti-aliasing. Let $\mathbf{x}(\mathbf{u})$ be the extracted low-resolution image at warped locations $\mathbf{p}'$:

$$\mathbf{x}(\mathbf{u}) \triangleq \mathcal{I}(\mathsf{w}(\{\mathbf{p}\}, \mathbf{u})). \tag{5}$$

Intuitively, $\mathbf{x}(\mathbf{u})$ is a patch extracted from $\mathcal{I}$ according to the shift, rotation and scale parameters of $\mathbf{u}$, as shown in Fig. 1, right panel. It is this patch of data that we seek to model generatively. Note that the dimensionality of $\mathbf{x}(\mathbf{u})$ is equal to the cardinality of $\{\mathbf{p}\}$, where $\{\mathbf{p}\}$ denotes the set of pixel coordinates of the canonical image $\mathbf{v}$. Unlike standard generative learning tasks, the data $\mathbf{x}(\mathbf{u})$ is not static but changes with the latent variables $\mathbf{u}$. Given $\mathbf{v}$ and $\mathbf{u}$, we model the top-down generative process over[2] $\mathbf{x}$ with a Gaussian distribution having a diagonal covariance matrix $\sigma^2 \mathrm{I}$:

$$p(\mathbf{x}|\mathbf{v}, \mathbf{u}, \mathcal{I}) \propto \exp\left( -\frac{1}{2} \sum_i \frac{(x_i(\mathbf{u}) - v_i)^2}{\sigma_i^2} \right). \tag{6}$$

The fact that we do not seek to model the rest of the regions/pixels of $\mathcal{I}$ is by design. By using 2D similarity transformation to mimic attention, we can discard the complex background of the scene and let the generative model focus on the object of interest. The proposed generative model takes the following form:

$$p(\mathbf{x}, \mathbf{v}, \mathbf{u}|\mathcal{I}) = p(\mathbf{x}|\mathbf{v}, \mathbf{u}, \mathcal{I})p(\mathbf{v})p(\mathbf{u}), \tag{7}$$

where for $p(\mathbf{u})$ we use a flat prior that is constant for all $\mathbf{u}$, and $p(\mathbf{v})$ is defined by a 2-layer Gaussian Deep Belief Network. The conditional $p(\mathbf{x}|\mathbf{v}, \mathbf{u}, \mathcal{I})$ is given by a Gaussian distribution as in Eq. 6. To simplify the inference procedure, $p(\mathbf{x}|\mathbf{v}, \mathbf{u}, \mathcal{I})$ and the GDBN model of $\mathbf{v}$, $p(\mathbf{v})$, will share the same noise parameters $\sigma_i$.

## 4 Inference

While the generative equations in the last section are straightforward and intuitive, inference in these models is typically intractable due to the complicated energy landscape of the posterior. During inference, we wish to compute the distribution over the gaze variables $\mathbf{u}$ and canonical object $\mathbf{v}$ given the big image $\mathcal{I}$. Unlike in standard RBMs and DBNs, there are no simplifying factorial assumptions about the conditional distribution of the latent variable $\mathbf{u}$. Having a 2D similarity transformation is reminiscent of third-order Boltzmann machines with $\mathbf{u}$ performing top-down multiplicative gating of the connections between $\mathbf{v}$ and $\mathcal{I}$. It is well known that inference in these higher-order models is rather complicated.

One way to perform inference in our model is to resort to Gibbs sampling by computing the set of alternating conditional posteriors: The conditional distribution over the canonical image $\mathbf{v}$ takes the following form:

$$p(\mathbf{v}|\mathbf{u}, \mathbf{h}^1, \mathcal{I}) = \mathcal{N}\left(\frac{\boldsymbol{\mu} + \mathbf{x}(\mathbf{u})}{2}; \boldsymbol{\sigma}^2\right), \tag{8}$$

where $\mu_i = b_i + \sigma_i^2 \sum_j W_{ij} h_j^1$ is the top-down influence of the DBN. Note that if we know the gaze variable $\mathbf{u}$ and the first layer of hidden variables $\mathbf{h}^1$, then $\mathbf{v}$ is simply defined by a Gaussian distribution, where the mean is given by the average of the top-down influence and bottom-up information from $\mathbf{x}$. The conditional distributions over $\mathbf{h}^1$ and $\mathbf{h}^2$ given $\mathbf{v}$ are given by the standard DBN inference equations [1]. The conditional posterior over the gaze variables $\mathbf{u}$ is given by:

$$p(\mathbf{u}|\mathbf{x}, \mathbf{v}) = \frac{p(\mathbf{x}|\mathbf{u}, \mathbf{v})p(\mathbf{u})}{p(\mathbf{x}|\mathbf{v})},$$

$$\log p(\mathbf{u}|\mathbf{x}, \mathbf{v}) \propto \log p(\mathbf{x}|\mathbf{u}, \mathbf{v}) + \log p(\mathbf{u}) = \frac{1}{2}\sum_i \frac{(x_i(\mathbf{u}) - v_i)^2}{\sigma_i^2} + const. \tag{9}$$

Using Bayes' rule, the unnormalized log probability of $p(\mathbf{u}|\mathbf{x}, \mathbf{v})$ is defined in Eq. 9. We stress that this equation is atypical in that the random variable of interest $\mathbf{u}$ actually affects the conditioning variable $\mathbf{x}$ (see Eq. 5) We can explore the gaze variables using Hamiltonian Monte Carlo (HMC) algorithm [27, 28]. Intuitively, conditioned on the canonical object $\mathbf{v}$ that our model has in "mind", HMC searches over the entire image $\mathcal{I}$ to find a region $\mathbf{x}$ with a good match to $\mathbf{v}$.

If the goal is only to find the MAP estimate of $p(\mathbf{u}|\mathbf{x}, \mathbf{v})$, then we may want to use second-order methods for optimizing $\mathbf{u}$. This would be equivalent to the Lucas-Kanade framework in computer vision, developed for image alignment [29]. However, HMC has the advantage of being a proper MCMC sampler that satisfies detailed balance and fits nicely with our probabilistic framework.

The HMC algorithm first specifies the Hamiltonian over the position variables $\mathbf{u}$ and auxiliary momentum variables $\mathbf{r}$: $\mathcal{H}(\mathbf{u}, \mathbf{r}) = U(\mathbf{u}) + K(\mathbf{r})$, where the potential function is defined by $U(\mathbf{u}) = \frac{1}{2}\sum_i \frac{(x_i(\mathbf{u}) - v_i)^2}{\sigma_i^2}$ and the kinetic energy function is given by $K(\mathbf{r}) = \frac{1}{2}\sum_i r_i^2$. The dynamics of the system is defined by:

$$\frac{\partial \mathbf{u}}{\partial t} = \mathbf{r}, \qquad \frac{\partial \mathbf{r}}{\partial t} = -\frac{\partial \mathcal{H}}{\partial \mathbf{u}} \tag{10}$$

$$\frac{\partial \mathcal{H}}{\partial \mathbf{u}} = \frac{(\mathbf{x}(\mathbf{u}) - \mathbf{v})}{\sigma^2}\frac{\partial \mathbf{x}(\mathbf{u})}{\partial \mathbf{u}}, \tag{11}$$

$$\frac{\partial \mathbf{x}}{\partial \mathbf{u}} = \frac{\partial \mathbf{x}}{\partial \mathsf{w}(\{\mathbf{p}\}, \mathbf{u})}\frac{\partial \mathsf{w}(\{\mathbf{p}\}, \mathbf{u})}{\partial \mathbf{u}} = \sum_i \frac{\partial x_i}{\partial \mathsf{w}(\mathbf{p}_i, \mathbf{u})}\frac{\partial \mathsf{w}(\mathbf{p}_i, \mathbf{u})}{\partial \mathbf{u}}. \tag{12}$$

Observe that Eq. 12 decomposes into sums over single coordinate positions $\mathbf{p}_i = [x\ y]^{\mathsf{T}}$. Let us denote $\mathbf{p}'_i = \mathsf{w}(\mathbf{p}_i, \mathbf{u})$ to be the coordinate $\mathbf{p}_i$ warped by $\mathbf{u}$. For the first term on the RHS of Eq. 12,

$$\frac{\partial x_i}{\partial \mathsf{w}(\mathbf{p}_i, \mathbf{u})} = \nabla I(\mathbf{p}'_i), \quad \text{(dimension 1 by 2 )} \tag{13}$$

where $\nabla I(\mathbf{p}'_i)$ denotes the sampling of the gradient images of $I$ at the warped location $\mathbf{p}_i$. For the second term on the RHS of Eq. 12, we note that we can re-write Eq. 4 as:

$$\begin{bmatrix} x' \\ y' \end{bmatrix} = \begin{bmatrix} x & -y & 1 & 0 \\ y & x & 0 & 1 \end{bmatrix}\begin{bmatrix} a \\ b \\ \triangle x \\ \triangle y \end{bmatrix} + \begin{bmatrix} x \\ y \end{bmatrix}, \tag{14}$$

giving us

$$\frac{\partial \mathsf{w}(\mathbf{p}_i, \mathbf{u})}{\partial \mathbf{u}} = \left[ \begin{array}{cccc} x & -y & 1 & 0 \\ y & x & 0 & 1 \end{array} \right]. \tag{15}$$

HMC simulates the discretized system by performing leap-frog updates of $\mathbf{u}$ and $\mathbf{r}$ using Eq. 10. Additional hyperparameters that need to be specified include the step size $\epsilon$, number of leap-frog steps, and the mass of the variables (see [28] for details).

### 4.1 Approximate Inference

HMC essentially performs gradient descent with momentum, therefore it is prone to getting stuck at local optimums. This is especially a problem for our task of finding the best transformation parameters. While the posterior over $\mathbf{u}$ should be unimodal near the optimum, many local minima exist away from the global optimum. For example, in Fig. 2(a), the big image $\mathcal{I}$ is enclosed by the blue box, and the canonical image $\mathbf{v}$ is enclosed by the green box. The current setting of $\mathbf{u}$ aligns together the wrong eyes. However, it is hard to move the green box to the left due to the local optima created by the dark intensities of the eye. Resampling the momentum variable every iteration in HMC does not help significantly because we are modeling real-valued images using a Gaussian distribution as the residual, leading to quadratic costs in the difference between $\mathbf{x}(\mathbf{u})$ and $\mathbf{v}$ (see Eq. 9). This makes the energy barriers between modes extremely high.

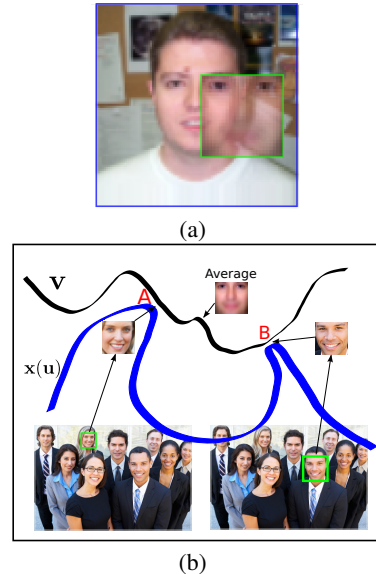

(a)

(b)

Figure 2: (a) HMC can easily get stuck at local optima. (b) Importance of modeling $p(\mathbf{u}|\mathbf{v}, \mathcal{I})$. Best in color.

To alleviate this problem we need to find good initializations of $\mathbf{u}$. We use a Convolutional Network (ConvNet) to perform efficient approximate inference, resulting in good initial guesses. Specifically, given $\mathbf{v}$, $\mathbf{u}$ and $\mathcal{I}$, we predict the change in $\mathbf{u}$ that will lead to the maximum $\log p(\mathbf{u}|\mathbf{x}, \mathbf{v})$. In other words, instead of using the gradient field for updating $\mathbf{u}$, we learn a ConvNet to output a better vector field in the space of $\mathbf{u}$. We used a fairly standard ConvNet architecture and the standard stochastic gradient descent learning procedure.

We note that standard feedforward face detectors seek to model $p(\mathbf{u}|\mathcal{I})$, while completely ignoring the canonical face $\mathbf{v}$. In contrast, here we take $\mathbf{v}$ into account as well. The ConvNet is used to initialize $\mathbf{u}$ for the HMC algorithm. This is important in a proper generative model because conditioning on $\mathbf{v}$ is appealing when multiple faces are present in the scene. Fig. 2(b) is a hypothesized Euclidean space of $\mathbf{v}$, where the black manifold represents canonical faces and the blue manifold represents cropped faces $\mathbf{x}(\mathbf{u})$. The blue manifold has a low intrinsic dimensionality of 4, spanned by $\mathbf{u}$. At A and B, the blue comes close to black manifold. This means that there are at least two modes in the posterior over $\mathbf{u}$. By conditioning on $\mathbf{v}$, we can narrow the posterior to a single mode, depending on whom we want to focus our attention. We demonstrate this exact capability in Sec. 6.3.

Fig. 3 demonstrates the iterative process of how approximate inference works in our model. Specifically, based on $\mathbf{u}$, the ConvNet takes a window patch around $\mathbf{x}(\mathbf{u})$ (72×72) and $\mathbf{v}$ (24×24) as input, and predicts the output $[\triangle x, \triangle y, \triangle \theta, \triangle s]$. In step 2, $\mathbf{u}$ is updated accordingly, followed by step 3 of alternating Gibbs updates of $\mathbf{v}$ and $\mathbf{h}$, as discussed in Sec. 4. The process is repeated. For the details of the ConvNet see the supplementary materials.

## 5 Learning

While inference in our framework localizes objects of interest and is akin to object detection, it is not the main objective. Our motivation is not to compete with state-of-the-art object detectors but rather propose a probabilistic generative framework capable of generative modeling of objects which are at unknown locations in big images. This is because labels are expensive to obtain and are often not available for images in an unconstrained environment.

To learn generatively without labels we propose a simple Monte Carlo based Expectation-Maximization algorithm. This algorithm is an unbiased estimator of the maximum likelihood objec-

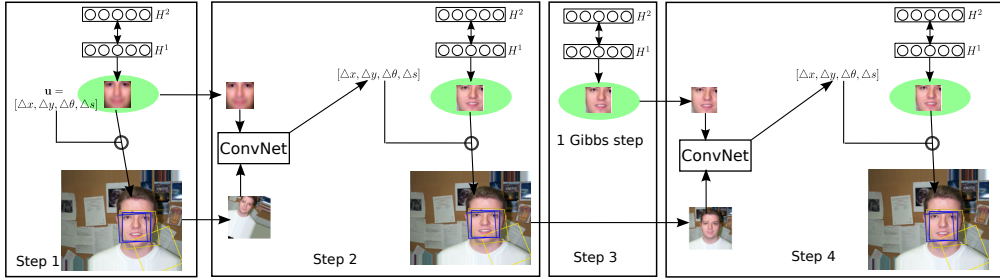

Figure 3: Inference process: **u** in step 1 is randomly initialized. The average **v** and the extracted **x(u)** form the input to a ConvNet for approximate inference, giving a new **u**. The new **u** is used to sample $p(\mathbf{v}|\mathcal{I}, \mathbf{u}, \mathbf{h})$. In step 3, one step of Gibbs sampling of the GDBN is performed. Step 4 repeats the approximate inference using the updated **v** and **x(u)**.

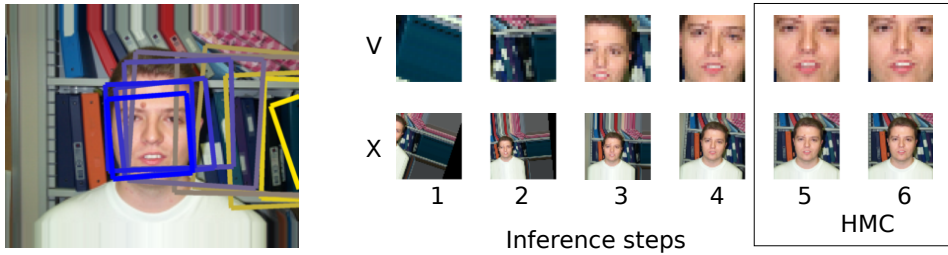

Figure 4: Example of an inference step. **v** is 24×24, **x** is 72×72. Approximate inference quickly finds a good initialization for **u**, while HMC provides further adjustments. Intermediate inference steps on the right are subsampled from 10 actual iterations.

tive. During the E-step, we use the Gibbs sampling algorithm developed in Sec. 4 to draw samples from the posterior over the latent gaze variables **u**, the canonical variables **v**, and the hidden variables $\mathbf{h}^1$, $\mathbf{h}^2$ of a Gaussian DBN model. During the M-step, we can update the weights of the Gaussian DBN by using the posterior samples as its training data. In addition, we can update the parameters of the ConvNet that performs approximate inference. Due to the fact that the first E-step requires a good inference algorithm, we need to pretrain the ConvNet using labeled gaze data as part of a bootstrap process. Obtaining training data for this initial phase is not a problem as we can jitter/rotate/scale to create data. In Sec. 6.2, we demonstrate the ability to learn a good generative model of face images from the CMU Multi-PIE dataset.

## 6 Experiments

We used two face datasets in our experiments. The first dataset is a frontal face dataset, called the Caltech Faces from 1999, collected by Markus Weber. In this dataset, there are 450 faces of 27 unique individuals under different lighting conditions, expressions, and backgrounds. We downsampled the images from their native 896 by 692 by a factor of 2. The dataset also contains manually labeled eyes and mouth coordinates, which will serve as the gaze labels. We also used the CMU Multi-PIE dataset [30], which contains 337 subjects, captured under 15 viewpoints and 19 illumination conditions in four recording sessions for a total of more than 750,000 images. We demonstrate our model's ability to perform approximate inference, to learn without labels, and to perform identity-based attention given an image with two people.

### 6.1 Approximate inference

We first investigate the critical inference algorithm of $p(\mathbf{u}|\mathbf{v}, \mathcal{I})$ on the Caltech Faces dataset. We run 4 steps of approximate inference detailed in Sec. 4.1 and diagrammed in Fig. 3, followed by three iterations of 20 leap-frog steps of HMC. Since we do not initially know the correct **v**, we initialize **v** to be the average face across all subjects.

Fig. 4 shows the image of **v** and **x** during inference for a test subject. The initial gaze box is colored yellow on the left. Subsequent gaze updates progress from yellow to blue. Once ConvNet-based approximate inference gives a good initialization, starting from step 5, five iterations of 20 leap-frog steps of HMC are used to sample from the the posterior.

Fig. 5 shows the quantitative results of Intersection over Union (IOU) of the ground truth face box and the inferred face box. The results show that inference is very robust to initialization and requires

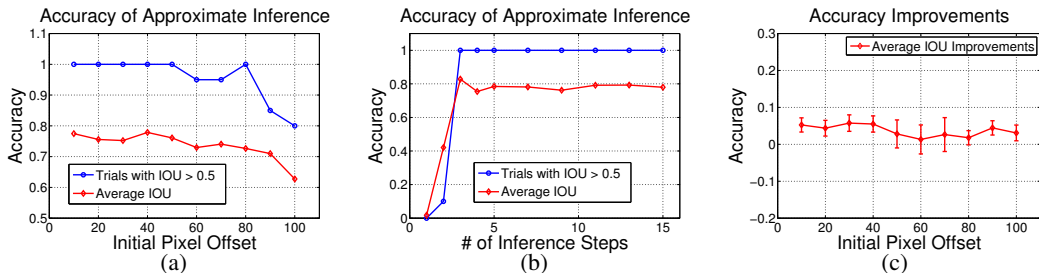

Figure 5: (a) Accuracy as a function of gaze initialization (pixel offset). Blue curve is the percentage success of at least 50% IOU. Red curve is the average IOU. (b) Accuracy as a function of the number of approximate inference steps when initializing 50 pixels away. (c) Accuracy improvements of HMC as a function of gaze initializations.

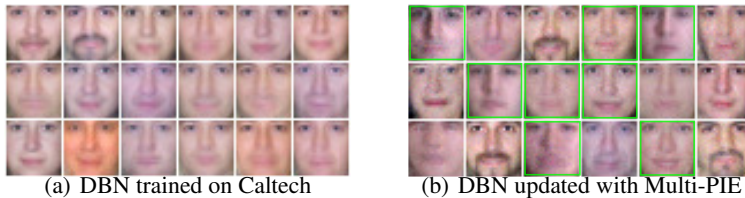

(a) DBN trained on Caltech      (b) DBN updated with Multi-PIE

Figure 6: **Left:** Samples from a 2-layer DBN trained on Caltech. **Right:** samples from an updated DBN after training on CMU Multi-PIE *without* labels. Samples highlighted in green are similar to faces from CMU.

only a few steps of approximate inference to converge. HMC clearly improves model performance, resulting in an IOU increase of about 5% for localization. This is impressive given that none of the test subjects were part of the training and the background is different from backgrounds in the training set.

|  | Our method | OpenCV | NCC | template |
|---|---|---|---|---|
| IOU > 0.5 | 97% | 97% | 93% | 78% |
| # evaluations | $O(c)$ | $O(whs)$ | $O(whs)$ | $O(whs)$ |

Table 1: Face localization accuracy. $w$: image width; $h$: image height; $s$: image scales; $c$: number of inference steps used.

We also compared our inference algorithm to the template matching in the task of face detection. We took the first 5 subjects as test subjects and the rest as training. We can localize with **97%** accuracy (IOU > 0.5) using our inference algorithm[3]. In comparison, a near state-of-the-art face detection system from OpenCV 2.4.9 obtains the same 97% accuracy. It uses Haar Cascades, which is a form of AdaBoost[4]. Normalized Cross Correlation [31] obtained 93% accuracy, while Euclidean distance template matching achieved an accuracy of only 78%. However, note that our algorithm looks at a constant number of windows while the other baselines are all based on scanning windows.

## 6.2 Generative learning without labels

| nats | No CMU training | CMU w/o labels | CMU w/ labels |
|---|---|---|---|
| Caltech Train | 617±0.4 | 627±0.5 | 569±0.6 |
| Caltech Valid | 512±1.1 | 503±1.8 | 494±1.7 |
| CMU Train | 96±0.8 | 499±0.1 | 594±0.5 |
| CMU Valid | 85±0.5 | 387±0.3 | 503±0.7 |
| $\log \hat{Z}$ | 454.6 | 687.8 | 694.2 |

Table 2: Variational lower-bound estimates on the log-density of the Gaussian DBNs (higher is better).

The main advantage of our model is that it can learn on large images of faces without localization label information (no manual cropping required). To demonstrate, we use both the Caltech and the CMU faces dataset. For the CMU faces, a subset of 2526 frontal faces with ground truth labels are used. We split the Caltech dataset into a training and a validation set. For the CMU faces, we first took 10% of the images as training cases for the ConvNet for approximate inference. This is needed due to the completely different backgrounds of the Caltech and CMU datasets. The remaining 90% of the CMU faces are split into a training and validation set. We first trained a GDBN with 1024 $\mathbf{h}^1$ and 256 $\mathbf{h}^2$ hidden units on the Caltech training set. We also trained

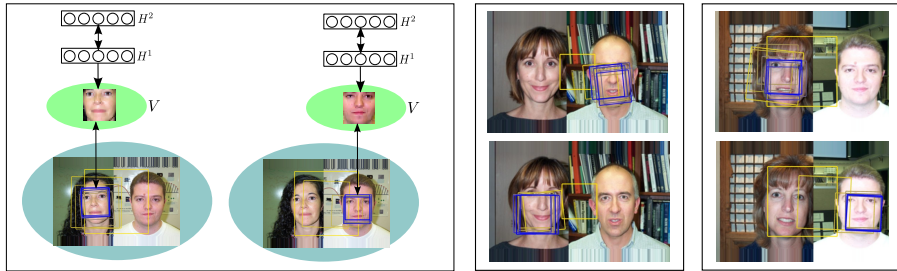

Figure 7: **Left:** Conditioned on different $\mathbf{v}$ will result in a different $\triangle\mathbf{u}$. Note that the initial $\mathbf{u}$ is exactly the same for two trials. **Right:** Additional examples. The only difference between the top and bottom panels is the conditioned $\mathbf{v}$. Best viewed in color.

a ConvNet for approximate inference using the Caltech training set and 10% of the CMU training images.

Table 2 shows the estimates of the variational lower-bounds on the average log-density (higher is better) that the GDBN models assign to the ground-truth cropped face images from the training/test sets under different scenarios. In the left column, the model is only trained on Caltech faces. Thus it gives very low probabilities to the CMU faces. Indeed, GDBNs achieve a variational lower-bound of only 85 nats per test image. In the middle column, we use our approximate inference to estimate the location of the CMU training faces and further trained the GDBN on the newly localized faces. This gives a dramatic increase of the model performance on the CMU Validation set[5], achieving a lower-bound of 387 nats per test image. The right column gives the best possible results if we can train with the CMU manual localization labels. In this case, GDBNs achieve a lower-bound of 503 nats. We used Annealed Importance Sampling (AIS) to estimate the partition function for the top-layer RBM. Details on estimating the variational lower bound are in the supplementary materials.

Fig. 6(a) further shows samples drawn from the Caltech trained DBN, whereas Fig. 6(b) shows samples after training with the CMU dataset using estimated $\mathbf{u}$. Observe that samples in Fig. 6(b) show a more diverse set of faces. We trained GDBNs using a greedy, layer-wise algorithm of [1]. For the top layer we use Fast Persistent Contrastive Divergence [32], which substantially improved generative performance of GDBNs (see supplementary material for more details).

### 6.3 Inference with ambiguity

Our attentional mechanism can also be useful when multiple objects/faces are present in the scene. Indeed, the posterior $p(\mathbf{u}|\mathbf{x}, \mathbf{v})$ is conditioned on $\mathbf{v}$, which means that *where to attend* is a function of the canonical object $\mathbf{v}$ the model has in "mind" (see Fig. 2(b)). To explore this, we first synthetically generate a dataset by concatenating together two faces from the Caltech dataset. We then train approximate inference ConvNet as in Sec. 4.1 and test on the held-out subjects. Indeed, as predicted, Fig. 7 shows that depending on which canonical image is conditioned, the same exact gaze initialization leads to two very different gaze shifts. Note that this phenomenon is observed across different scales and location of the initial gaze. For example, in Fig. 7, right-bottom panel, the initialized yellow box is mostly on the female's face to the left, but because the conditioned canonical face $\mathbf{v}$ is that of the right male, attention is shifted to the right.

## 7 Conclusion

In this paper we have proposed a probabilistic graphical model framework for learning generative models using attention. Experiments on face modeling have shown that ConvNet based approximate inference combined with HMC sampling is sufficient to explore the complicated posterior distribution. More importantly, we can generatively learn objects of interest from novel big images. Future work will include experimenting with faces as well as other objects in a large scene. Currently the ConvNet approximate inference is trained in a supervised manner, but reinforcement learning could also be used instead.

## Acknowledgements

The authors gratefully acknowledge the support and generosity from Samsung, Google, and ONR grant N00014-14-1-0232.

## Footnotes

[1]Other generative models can also be used with our attention framework.

[2]We will often omit dependence of $\mathbf{x}$ on $\mathbf{u}$ for clarity of presentation.

[3]$\mathbf{u}$ is randomly initialized at $\pm$ 30 pixels, scale range from 0.5 to 1.5.

[4]OpenCV detection uses pretrained model from haarcascade_frontalface_default.xml, scaleFactor=1.1, minNeighbors=3 and minSize=30.

[5]We note that we still made use of labels coming from the 10% of CMU Multi-PIE training set in order to pretrain our ConvNet. "w/o labels" here means that no labels for the CMU Train/Valid images are given.

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
