[Supplementary Material]

# Learning Generative Models with Visual Attention - Supplementary Materials

Yichuan Tang,  Nitish Srivastava,  Ruslan Salakhutdinov

Department of Computer Science
University of Toronto
Toronto, Ontario, Canada
{tang,nitish,rsalakhu}@cs.toronto.edu

## Convolutional Neural Network

The training of ConvNet for approximate inference is standard and did not involve any special 'tricks'. We used SGD with minibatch size of 128 samples. We used a standard ConvNet architecture with convolution C layers followed by max-pooling S layers. The ConvNet takes as input $\mathbf{x}$ and $\mathbf{v}$ to predict change in $\mathbf{u}$ such that to maximize $\log p(\mathbf{u}|\mathbf{x}, \mathbf{v})$. In order to better predict change of $\mathbf{u}$, $\mathbf{x}$ as well as a bigger border around $\mathbf{x}$ are used as the input to the ConvNet. Therefore, $\mathbf{x}$ has resolution 72×72 and $\mathbf{v}$ has resolution of 24×24.

Two handle two different inputs with different resolutions, two different "streams" are used in this ConvNet architecture. One stream will process $\mathbf{x}$ and another one for $\mathbf{v}$. These two streams will be combined multiplicatively after subsampling the $\mathbf{x}$ stream by a factor of 3. The rest of the ConvNet is same as the standard classification ConvNets, except that we use mean squared error as our cost function. See Figure 1 for a visual diagram of what the convolutional neural network architecture used.

| layer | type | latent variables | filter size | # weights |
|---|---|---|---|---|
| 0 | input $\mathbf{x}$ | maps:3 72x72 | - | - |
| 1 | input $\mathbf{v}$ | maps:3 24x24 | - | - |
| 2 | Conv of layer 0 | maps:16 66x66 | 7x7 | 2352 |
| 3 | Pooling | maps:16 22x22 | 3x3 | - |
| 4 | Conv of layer 1 | maps:16 22x22 | 5x5 | 1200 |
| 5 | Combine layers 3,4 | maps:16 22x22 | - | - |
| 6 | Fully connected | 1024 | - | 7.9M |
| 7 | Fully connected | 4 | - | 4K |

Table 1: Model architectures of the convolutional neural network used during approximate inference.

Table 1 details the exact model architecture used. In layer 5, the two streams have the same number of hidden maps and hidden topography. We combine these two multiplicatively by multiplying their activation element-wise. This creates a third-order flavor and is more powerful for the task of determining where to shift attention to next.

## Gaussian Deep Belief Network

The training of the Gaussian Deep Belief Network is performed in a standard greedy layerwise fashion. The first layer Gaussian Restricted Boltzmann Machine is trained with the Contrastive Divergence algorithm where the standard deviation of each visible unit is learned as well. After training

Figure 1: A visual diagram of the convolutional net used for approximate inference.

the first layer, we use Eq. 3 to obtain first hidden layer binary probabilities. We then train a 2nd binary-binary Restricted Boltzmann Machine using the fast persistent contrastive divergence learning algorithm. This greedy training leads us to the Gaussian Deep Belief Network. No finetuning of the entire network is performed.

**Quantitative evaluation for Gaussian Deep Belief Network**

For quantitative evaluation, we approximate the standard variational lower bound on the log likelihood of the Gaussian Deep Belief Network. The model is a directed model:

$$p(\mathbf{v}, \mathbf{h}^1, \mathbf{h}^2) = p(\mathbf{v}|\mathbf{h}^1)p(\mathbf{h}^1, \mathbf{h}^2) \tag{1}$$

For any approximating posterior distribution $q(\mathbf{h}^1|\mathbf{v})$, the GDBN's log-likelihood has this lower variational bound:

$$\log \sum_{\mathbf{h}^1} p(\mathbf{v}, \mathbf{h}^1) \geq \sum_{\mathbf{h}^1} q(\mathbf{h}^1|\mathbf{v})[\log p(\mathbf{v}|\mathbf{h}^1) + \log p^*(\mathbf{h}^1)] - \log Z + \mathcal{H}(q(\mathbf{h}^1|\mathbf{v})) \tag{2}$$

The entropy $\mathcal{H}(q(\mathbf{h}^1|\mathbf{v}))$ can be calculated since we made the factorial assumption on $q(\mathbf{h}^1|\mathbf{v})$.

$$\log p(\mathbf{v}|\mathbf{h}^1) = -\sum \log \sigma_i - \frac{D}{2}\log 2\pi - \frac{1}{2}\sum_i^D \frac{(x-\mu)^2}{\sigma_i^2} \tag{3}$$

$$\log p^*(\mathbf{h}^1) = \mathbf{b}^\mathsf{T}\mathbf{h}^1 + \log \sum_j \exp\{\mathbf{h}^1\mathbf{W}_j + c_j\} \tag{4}$$

In order to calculate the expectation of the approximating posteriors, we use Monte Carlo sampling.

$$\sum_{\mathbf{h}^1} q(\mathbf{h}^1|\mathbf{v}) \log p^*(\mathbf{v}, \mathbf{h}^1) \approx \frac{1}{M}\sum_{m=1}^M \log p^*(\mathbf{v}, \mathbf{h}^{1(m)}) \tag{5}$$

$$= -\sum \log \sigma_i - \frac{D}{2}\log 2\pi - \frac{1}{2}\sum_i^D \frac{(x-\mu)^2}{\sigma_i^2} \tag{6}$$

$$+ \mathbf{b}^\mathsf{T}\mathbf{h}^1 + \log \sum_j \exp\{\mathbf{h}^1\mathbf{W}_j + c_j\} \tag{7}$$

where $\mathbf{h}^{1(m)}$ is the $m$-th sample from the posterior $q(\mathbf{h}^1|\mathbf{v})$.

In order to calculate the partition function of the top-most layer of the GDBN, we use Annealed Importance Sampling (AIS). We used 100 chains with 50,000 intermediate distributions to estimate the partition function of the binary-binary RBM which forms the top layer of the GDBN. Even though AIS is an unbiased estimator of the partition function, it is prone to under-estimating it due to bad mixing of the chains. This causes the log probability to be over-estimated. Therefore the variational lower bounds reported in our paper are not strictly guaranteed to be lower bounds and are subject to errors. However, we believe that the margin of error is unlikely to be high enough to affect our conclusions.

## Additional Results

We present some more examples of inference process of our framework.

Figure 2: Example of an approximate inference steps. $\mathbf{v}$ is 24×24, $\mathbf{x}$ is 72×72. Approximate inference quickly finds a good initialization for $\mathbf{u}$, while HMC makes small adjustments.

Below, we show some success cases and a failure case for inference on the CMU Multi-PIE dataset. The initial gaze variables of $\mathbf{u}$ are highlighted in yellow and later iterations are highlighted with color gradually changing to blue.

(a) Ex. 1    (b) Ex. 2    (c) Ex. 3    (d) Ex. 4

Figure 3: E-step for learning on CMU Multi-PIE. (a),(b),(c) are successful. (d) is a failure case.