[Reviews · NeurIPS 2014]

Submitted by Assigned_Reviewer_2

Summary:

This paper proposes a deep learning based spatial attention mechanism based on a probabilistic generative model. The approach enables identifying novel objects in large images, and hence might allow for better exploitation of unlabeled, uncropped data.

Main Comments:

This paper takes a solid step towards being able to apply deep learning methods to large uncropped, unlabeled images. The motivation is compelling: previous DL approaches have required ‘curated’ training data that sticks the object essentially in the center with minimal clutter/occlusions—overcoming this limitation would truly enable learning from unlimited unlabeled data.

The experimental section contains a number of interesting experiments verifying that the approximate inference method is working, and that HMC is still worthwhile beyond this. The network can perform some notable and rarely addressed tasks such as the ability to learn generative models from large images without labels, and the ability to shift attention by conditioning on various target stimuli.

The paper claims that the algorithm runs in time O(1) given the size of the image. However initializations farther from the target image are likely to require more approximate inference steps, so there is an implicit dependence on image size. It could be interesting to plot the required number of steps as a function of image size to see this scaling. Also the window patch supplied to the conv net might also need to be increased for large images, so the scaling is not really O(1).

The paper is clearly written and easy to follow.
Summary: This paper takes a solid step towards being able to learn deep network models using large uncropped, uncentered, unlabeled images—which could give access to virtually unlimited unlabeled data.

Submitted by Assigned_Reviewer_3

The paper proposes a generative model for images that uses an attentional mechanism to focus on items (objects) of interest within an image, and devoting the object model just to these attended objects, as opposed to the entire image (as in most deep network approaches). Latent variable representations for the object and its pose (position, size and orientation) are inferred from the image using hamiltonian monte carlo. Performance is demonstrated on the Caltech and CMU-PIE face datasets, showing good performance of the model.

Overall I love this paper. It proposes something that is totally sensible and long overdue in both discriminative and generative models of images - i.e., employing an attentional mechanism. I feel that this is an important advance, and although many aspects of the work could probably be expanded and improved, I think it will generate strong interest at NIPS.

One area where I think this work could be improved is in learning the transformation model rather than assuming affine translation, scale and rotation. For example 2D projections of 3D objects will generate a richer set of warps/transformations. Also how to deal with occlusion?
Summary: Great paper - a very sensible approach with good results.

Submitted by Assigned_Reviewer_11

The goal of this study is to learn generative models of sensory signals in the attended region. The authors proposed a model that could successfully learn a generative model of frontal faces from large-size images that include faces among distractors.

The manuscript is well written overall. A wide range of researches are surveyed and combined, from neuroscience to computer vision to machine learning. I think the manuscript would be informative and useful to a broad audience.

The model itself is formulated in a generic way so that it could potentially be applied to a wide range of data. I think, however, the model is based on a couple of key assumptions that significantly limit its applicability. Namely, the model assumes “canonical image” (variable “v”), and also assumes similarity transformation. Therefore, the proposed approach could build a data generative model of “attended image regions”, but not the underlying, three-dimensional “objects” whose observations by sensors involve more than just similarity transformation. I think this is the reason why the section 6: experiments only examined frontal faces, not those from different views, nor other objects. This manuscript can be improved if such limitations towards the grand goal are clearly discussed.
Summary: An interesting approach to a challenging problem of “generative model of attended image regions”, by combining several strands of prior researches from different areas. Thought-provoking paper, but I think the proposed model is still fairly limited.
Author Feedback
Author rebuttal: We thank the reviewers for their valuable feedback, and will incorporate various suggestions for the final submission.

Reviewer_11:
We have been thinking about the possibility of extending our framework to extracting underlying 3-D objects from 2D images, as well as applying our model to other object classes. We will add a discussion on this point.

Reviewer_2:
We agree with your comment regarding O(1) and will include a plot of # of inference steps versus initial position.

Reviewer_3:
Going beyond simple affine transformations as well as occlusion handling is certainly part of the plan for our future work. For example, using 3rd-order pixel-wise gating, similar to the model proposed by "Robust Boltzmann Machines for Recognition and Denoising (CVPR 12)", could be incorporated into the existing framework to deal with some basic occlusions.